# Could perturbed fetal development of the ovary contribute to the development of polycystic ovary syndrome in later life?

Monica D. Hartanti[1,2], Roseanne Rosario[3], Katja Hummitzsch[1], Nicole A. Bastian[1], Nicholas Hatzirodos[1], Wendy M. Bonner[1], Rosemary A. Bayne[3], Helen F. Irving-Rodgers[1,4], Richard A. Anderson[3], Raymond J. Rodgers[1] *

1 Discipline of Obstetrics and Gynaecology, School of Medicine, Robinson Research Institute, The University of Adelaide, Adelaide, SA, Australia, 2 Faculty of Medicine, Trisakti University, Jakarta, Indonesia, 3 Medical Research Council Centre for Reproductive Health, University of Edinburgh, Edinburgh, United Kingdom, 4 School of Medical Science, Griffith University, QLD, Australia

* ray.rodgers@adelaide.edu.au

**Data Availability Statement:** All relevant data are within the paper and its Supporting Information files.

## Abstract

Polycystic ovary syndrome (PCOS) affects around 10% of young women, with adverse consequences on fertility and cardiometabolic outcomes. PCOS appears to result from a genetic predisposition interacting with developmental events during fetal or perinatal life. We hypothesised that PCOS candidate genes might be expressed in the fetal ovary when the stroma develops; mechanistically linking the genetics, fetal origins and adult ovarian phenotype of PCOS. In bovine fetal ovaries (n = 37) of 18 PCOS candidate genes only *SUMO1P1* was not expressed. Three patterns of expression were observed: early gestation (*FBN3*, *GATA4*, *HMGA2*, *TOX3*, *DENND1A*, *LHCGR* and *FSHB*), late gestation (*INSR*, *FSHR*, and *LHCGR*) and throughout gestation (*THADA*, *ERBB4*, *RAD50*, *C8H9orf3*, *YAP1*, *RAB5B*, *SUOX* and *KRR1*). A splice variant of *FSHB* exon 3 was also detected early in the bovine ovaries, but exon 2 was not detected. Three other genes, likely to be related to the PCOS aetiology (*AMH*, *AR* and *TGFB1I1*), were also expressed late in gestation. Significantly within each of the three gene groups, the mRNA levels of many genes were highly correlated with each other, despite, in some instances, being expressed in different cell types. TGFβ is a well-known stimulator of stromal cell replication and collagen synthesis and TGFβ treatment of cultured fetal ovarian stromal cells inhibited the expression of *INSR*, *AR*, *C8H9orf3* and *RAD50* and stimulated the expression of *TGFB1I1*. In human ovaries (n = 15, < 150 days gestation) many of the same genes as in bovine (*FBN3*, *GATA4*, *HMGA2*, *FSHR*, *DENND1A* and *LHCGR* but not *TOX3* or *FSHB*) were expressed and correlated with each other. With so many relationships between PCOS candidate genes during development of the fetal ovary, including TGFβ and androgen signalling, we suggest that future studies should determine if perturbations of these genes in the fetal ovary can lead to PCOS in later life.

**Funding:** The research activities conducted in Australia were funded by the Australia Awards Scholarship from the Australian Government, The University of Adelaide, the National Health and Medical Research Council of Australia, the NHMRC Centre for Research Excellence in the Evaluation, Management and Health Care Needs of Polycystic Ovary Syndrome. The research activities conducted in the UK were supported by the UK Medical Research Council (grant no G1100357).

**Competing interests:** The authors have declared that no competing interests exist.

## Introduction

Polycystic ovary syndrome (PCOS) is a reproductive and metabolic condition with psychological consequences [1]. It affects about one in ten women, significantly impacting their health and well-being [2,3]. Women with PCOS suffer symptoms of excess androgen (hirsutism, acne, central adiposity), reproductive dysfunction (infertility, menstrual irregularity, miscarriage, pregnancy complications) and metabolic complications [4,5]. Metabolic features include insulin resistance [6], compensatory hyperinsulinaemia and associated risk of obesity [4], gestational diabetes, impaired glucose tolerance, type 2 diabetes, non-alcoholic fatty liver disease, dyslipidaemia, and increased risk factors for cerebro- and cardiovascular disease [4,5,7,8]. Women with PCOS have a four-fold elevated risk of developing type 2 diabetes, irrespective of their BMI, and thus PCOS accounts for a quarter of type 2 diabetes in women of reproductive age [9]. Ultimately, anxiety and depression are increased and quality of life is reduced [10,11].

The cause(s) and aetiology of PCOS are not completely understood. PCOS appears to have a genetic origin [12–14]. Recent genome-wide association cohort studies (GWAS) into PCOS discovered new susceptibility loci associated with 17 genes [reviewed by [15]]: DENN domain-containing 1A *(DENND1A)*, luteinising hormone/chorionic gonadotrophin receptor *(LHCGR)*, follicle stimulating hormone beta subunit *(FSHB)*, follicle stimulating hormone receptor *(FSHR)*, yes associated protein 1 *(YAP1)*, insulin receptor *(INSR)*, ras-related protein *RAB5B*, TOX high mobility group box family member 3 *(TOX3)*, high mobility group AT-hook 2 *(HMGA2)*, chromosome 9 open reading frame 3 *(C9orf3)*, GATA binding protein 4 *(GATA4)*, Erb-B2 receptor tyrosine kinase 4 *(ERBB4)*, DNA repair protein *(RAD50)*, thyroid adenoma associated *(THADA)*, sulphite oxidase *(SUOX)*, Ca2+/calmodulin-dependent protein kinase *(KRR1)* and the small ubiquitin-like modifier 1 pseudogene 1 *(SUMO1P1)* [16–19]. Many human studies have been conducted in an effort to investigate these PCOS candidate genes, particularly *DENND1A* and its variants [20], *TOX3* [21], *FSHR* [22], *LHCGR* [23] and *INSR* [24]. Another PCOS susceptibility loci was identified earlier by case-cohort studies using microsatellite analyses and it is located in an intron of fibrillin 3 *(FBN3)* [25]. The contribution of these genes and the candidate gene loci in the development of the PCOS phenotype remain unclear. This is not an uncommon situation for many polygenic diseases, where identifying how a SNP or loci identified from genomic studies are related to the aetiology of the disease can take considerable research effort to succeed [26].

Several observations also support the concept that a predisposition to PCOS develops in fetal or neonatal life. For instance in human babies, low ponderal index (kg/m$^3$) is associated with the presence of all three PCOS symptoms (menstrual dysfunction, hyperandrogenism and polycystic ovaries) in later life [27]. Similarly, women born with congenital adrenal hyperplasia, which causes elevated androgen exposure in fetal life, are also more likely to develop PCOS [28]. Animal models of PCOS commonly involve treating pregnant mothers (sheep, monkeys) or newborns (mice and rats) with androgens, with the resulting offspring showing features of PCOS when they reach adulthood [29–32]. In the absence of any obvious environmental androgens that might cause PCOS by acting on the developing fetus, this area was further advanced when it was discovered that pregnant women with PCOS had elevated anti-Mullerian hormone (AMH) levels [33]. The concept that these elevated circulating levels of AMH in the PCOS mothers could elevate level of androgens in the fetus was considered as a cause of PCOS [33]. However, this concept currently awaits proof in humans.

Some of the cardinal features of the PCOS ovary are the increased production of androgens, the presence of many antral follicles (inaccurately referred to as cysts in the name PCOS) and substantially more fibrous collagen-rich stroma and tunica albuginea [34]. How or why the stroma is different is not known. Obvious possibilities include a fibrosis occurring in the adult

PCOS ovary or an over production of ovarian stroma when it first develops. Ovarian stroma first arises by migration or penetration from the mesonephros underlying the gonadal ridge [35]; this is a consistent process observed in ovine [36], bovine [35,37] and human [38,39] ovarian development.

The idea of a relationship between the genetics, the fetal origins and the fibrous ovarian phenotype in adult women, first became apparent when it was discovered that *FBN3* is expressed in human and bovine ovarian stroma only in the early stage of fetal development when the stroma penetrates the ovary from the mesonephros [40]. Fibrillins regulate TGFβ activity in tissues by their ability to bind latent TGFβ binding proteins [41] and TGFβ is well known to stimulate stromal growth and collagen deposition [42]. Thus the concept was developed that increased TGFβ bioactivity during fetal development could contribute to the elevated amounts of stroma in PCOS ovaries [40,43]. These developmental changes might be subject to environmental modification, during development or postnatally [44].

Further impetus for these concepts followed when it was found that *LHCGR*, another PCOS gene identified by GWAS, was also expressed early in the fetal ovary and its expression correlated with expression of *FBN3* [45]. To date the potential expression of other PCOS genes identified by GWAS has not been investigated in fetal ovaries. Access to human fetal ovaries is relatively limited especially at later gestational ages and so to examine the expression of other PCOS genes identified by GWAS, we additionally chose an animal model. The histology and development of the bovine fetal ovary is similar to the human fetal ovary, as is the length of gestation [46,47]. Therefore, we analysed the expression of PCOS candidate genes identified by GWAS and additional genes, including androgen receptor *(AR)*, transforming growth factor beta 1 induced transcript 1 (*TGFB1I1*; also known as Hic5), *FBN3* and *AMH*, in bovine fetal ovaries throughout gestation as well as in adult ovaries. We examined the ovarian location of expression of these genes by laser capture micro-dissection and examined their expression in bovine fetal ovarian stromal cells *in vitro*. We also analysed the expression of some PCOS candidate genes identified by GWAS in human fetal ovaries from the first half of gestation.

## Materials and methods

### Ethics approvals

The human study was approved by the Lothian Research Ethics Committee (ref 08/S1101/1) and all methods were performed in accordance with the relevant guidelines and regulations of this approval. For our bovine study, there were no ethical issues regarding this project since fetal bovine ovaries were collected from the local abattoir which was processing animals for the food chain.

### Collection of human fetal ovaries

Human fetal ovaries (n = 15, 8–20 weeks of gestation) were obtained following medical termination of pregnancy as previously described [48]. Pregnancies were all terminated for social reasons and all fetuses appeared morphologically normal. Maternal informed consent was obtained and the study was approved by the Lothian Research Ethics Committee (ref 08/S1101/1). To determine the gestational age, an ultrasound scan was performed and the length of the fetal foot was measured [49].

### Collection of bovine fetal and adult ovaries

Thirty-seven ovaries from fetuses of *Bos taurus* cows across gestation as well as five adult ovaries taken from non-pregnant animals across the oestrous cycle (early and mid-luteal phases

and the follicular phases) were collected from local abattoirs (Thomas Foods International, Murray Bridge, SA, Australia and Midfield Meat International, Warrnambool, Victoria, Australia) and were either transferred on ice in Hank's Balanced-Salt solution containing $Mg^{2+}$ and $Ca^{2+}$ ($HBSS^{+/+}$, Sigma-Aldrich Pty Ltd, Castle Hill, NSW, Australia) to the laboratory or dissected and processed on site. To estimate the gestational age of fetal samples, the crown-rump length (CRL) was measured [50].

For laser capture micro-dissection (LCM), fetal ovaries from early (n = 5, 10–17 weeks of gestation) and late (n = 5, 36–39 weeks of gestation) stage of development were embedded into cryomolds filled with optimal cutting temperature (OCT) compound (ProSciTech, Thuringowa Central QLD, Australia) with the hilum on the side of the mould, frozen on dry ice and stored at -80˚C. For RNA extraction from the ovarian samples, 27 whole fetal and 1–3 pieces of cortical area containing preantral follicles and stroma from 5 adult ovaries were snap-frozen on dry ice and stored at -80˚C freezer for further analysis, whereas for the laser capture micro-dissected samples the cortical and medullar area were dissected from the fetal ovary and subsequently used for qRT-PCR.

## Sex determination of bovine fetuses

Genomic DNA was extracted from the tail of fetuses with a CRL less than 10 cm using the Wizard SV Genomic DNA Purification System (Promega Australia, Alexandria, NSW, Australia) according to the manufacturer's instructions and subsequently amplified. Two pairs of primers specific for a region in the SRY-determining sequence (sense primer: 5′ –TCACTCCTGCAAA AGGAGCA–3′, antisense primer: 5′ –TTATTGTGGCCCAGGCTTG–3′) and for the 18S ribosomal RNA (18S) gene sequence were used for amplifying the genomic DNA in individual reactions. Sex determining region Y (SRY) product sequences were verified using a PCR as previously described [35].

## Classification of bovine fetal ovaries

Non-laser capture micro-dissected fetal ovary samples were grouped into five groups based on their histological morphology; stage I: ovigerous cord formation (n = 7, 79 ± 6 days of gestation), stage II: ovigerous cord breakdown (n = 4, 127 ± 6 days), stage III: follicle formation (n = 3, 173 ± 12 days), stage IV: ovarian surface epithelium formation (n = 8, 234 ± 9 days) and stage V: tunica albuginea formation (n = 5, 264 ± 6 days) [37].

## Laser capture micro-dissection of bovine fetal ovaries

Serial frozen sections of 8 μm thickness were cut using a CM1800 cryostat (Leica Microsystems, Buffalo, IL). Cryosections were transferred onto room temperature PET (polyethylene terephthalate) membrane frame slides (Leica Microsystems, North Ryde, NSW, Australia) and stored in a RNase-free slide box. Slides were fixed in 70% ethanol in DEPC-treated water, stained in 1% cresyl violet acetate (pH 7.75) (ProSciTech) in 70% ethanol [51], washed for 30 sec with 70%, 90%, and 100% ethanol, followed by 1 min in 100% ethanol and either dissected directly or stored overnight at -80˚C in a parafilm-sealed 50 ml tube with dry desiccant. Slides were transferred to the LCM room on dry ice, then incubated for 5 min at 55˚C immediately prior to LCM (LMD AS, Leica Microsystems, North Ryde, NSW, Australia). Excised cortical and medullar samples [S1 Fig [52]] were collected into 20 μl of Ambion lysis solution for RNA isolation (Invitrogen, Carlsbad, CA, USA). Ten sections per area collected from 5 animals for each early and late stage of development were used for RNA isolation and subsequent qRT-PCR analysis.

## RNA extraction and cDNA synthesis for human and bovine fetal ovaries

For human samples, RNA from fetal ovaries was extracted using the RNeasy Micro Kit (Qiagen, Crawley, UK) according to manufacturer's instructions. 500ng of RNA was reverse transcribed to cDNA using concentrated random primers and Superscript III reverse transcriptase (Life Technologies) according to manufacturer's instructions, and the cDNA synthesis reaction was diluted 1:20 before proceeding.

Whole fetal and adult bovine ovaries, as well as bovine fetal fibroblasts, were homogenised in 1 ml Trizol® (Thermo Fisher Scientific, Waltham, MA, USA) using the Mo Bio Powerlyser 24 (Mo Bio Laboratories Inc., Carlsbad, CA, USA) and RNA extracted according to manufacturer's instructions. The RNA concentration was then determined using a Nanodrop spectrophotometer (NanoDrop 1000 3.7.1, Nanodrop Technologies, Wilmington, DE, USA) based on the 260λ (wavelength) absorbance. Only samples which had a 260/280λ absorbance ratio > 1.8 were used and subsequently treated with DNase I (Promega/Life Technologies Australia Pty Ltd, Tullmarine, Vic, Australia) for 20 min at 37˚C.

For the laser capture micro-dissected samples, total RNA from tissue sections was extracted and DNase treated using the RNAqueous®-Micro Kit (Cat# AM1931, Thermo Fisher Scientific, Waltham, MA, USA) procedure for LCM according to the manufacturer's protocol. The integrity of RNA was assessed using the Experion™ automated electrophoresis system with the Experion RNA HighSens Analysis Kit (Cat# 7007105, Biorad, Hercules, California, USA). Only samples which had a RNA Quality Index (RQI) more than 4 were concentrated using 3M sodium acetate and 100% ethanol. Concentrated RNA samples were then reassessed using the Experion™ automated electrophoresis system and only those which had RQI more than 5 were used for qRT-PCR analysis.

Complementary DNA was then synthesised from 9–200 ng of DNase-treated RNA using 250 ng/μl random hexamers (Sigma, Adelaide, SA, Australia) and 200 U Superscript Reverse Transcriptase III (Thermo Fisher Scientific, Waltham, MA, USA) as previously described [53]. To exclude genomic contamination, negative control was generated by adding DEPC-water instead of the Superscript Reverse Transcriptase III.

## Quantitative real-time PCR

For each gene a standard curve of Cycle threshold (Ct) versus cDNA concentrations was then generated to test the combination of primers. The reactions were performed in duplicate using the following steps. cDNA dilutions were prepared in 10 μl reactions containing 1–2 μl of the 0.5 ng/μl cDNA dilution, 5 μl of Power SYBR™ Green PCR Master Mix (Applied Biosystems, Foster City, CA, USA), 0.1–0.3 μl each of forward and reserve primers [Sigma; S1 Table [52]] for the target genes, and 2.6–3.6 μl of DEPC-treated water. The amplification conditions were 95˚C for 15 s, then 60˚C for 60 s for 40 cycles using a Rotor-Gene 6000 series 1.7 thermal cycler (Qiagen GmbH, Hilden, Germany). Ct values were then determined using the Rotor-Gene 6000 software (Q series, Qiagen GmbH, Hilden, Germany) at a threshold of 0.05 normalised fluorescence units. Gene expression was determined by the mean of $2^{-\Delta Ct}$, where $\Delta Ct$ represents the target gene Ct– average of ribosomal protein L32 *(RPL32)* and peptidylprolyl isomerase A *(PPIA)* Ct for fetal samples and *RPL19* and *RPL32* for comparison of fetal and adult samples. These combinations of housekeeping genes were used because they were the most stable across all samples out of *RPL32*, *RPL19*, *PPIA*, actin beta *(ACTB)* and glyceraldehyde-3-phosphate dehydrogenase *(GAPDH)*.

## Primers for quantitative real-time PCR

For human fetal ovary samples, qRT-PCR primers were designed to amplify all transcript variants and to span exons [S1 Table [52]]. Primer pair efficiencies were calculated with the LinReg

PCR applet [54]. Each reaction was performed in a final volume of 10 μL, with 1x Brilliant III SYBR Green qPCR Master Mix (Agilent, Santa Clara, California), 20 pmol of each primer and 2 μL of diluted cDNA. Each cDNA sample was analysed in triplicate. For expression analyses in human fetal ovaries, target genes were normalised to the geometric mean expression of beta-2-microglobulin *(B2M)* and *RPL32*. Data analysis for relative quantification of gene expression and calculation of standard deviations was performed as outlined by [55].

For bovine samples, qRT-PCR primers were designed based on the published reference RNA sequences available in NCBI [S1 Table [52]] using Primer3 plus [56] and Net primer (PREMIER Biosoft Palo Alto, CA, USA) software. *DENND1A* is alternatively spliced and variant 1 is the full length mRNA and shares a number of exons in common with other splice variants. Primers used in the human (exons 21 and 22) detected *DENND1A.V1* but could also possibly detect variants 3 and 4 (*DENND1A.V3,4*) [S2–S4 Figs [52]] and in bovine (exon 12), the primers could also possibly detect the predicted variants 1, 2, 3 and 4 (*DENND1A.X1-4*) [S2–S4 Figs [52]]. To attempt to detect variant 2 in the bovine, primers to exon 20, and to a what is listed as intronic but corresponding to exon 21 in the human, were used [S4 Fig [52]]. The alignment of the human and bovine *DENND1A* sequences and the primer sequences was analysed with the T-coffee method for multiple sequences alignment [57]. Gene *C8H9orf3* is the bovine homologue of the human PCOS candidate gene *C9orf3*.

## Statistical analyses

All statistical analyses were carried out using Microsoft Office Excel 2013 (Microsoft Redmond, WA, USA) and GraphPad Prism version 7.00 (GraphPad Software Inc., La Jolla, CA, USA). All $2^{-\Delta Ct}$ data for each fetal ovarian sample were plotted in scatter plots and bar graphs to describe their levels of expression during ovarian development. To analyse the difference between the level of mRNA expression of each gene, one-way ANOVA with Holm-Sidak and Dunnet's *post-hoc* test were used for the whole ovary samples and fetal fibroblasts, respectively. For laser capture micro-dissected samples, unpaired t-test was applied. Pearson correlation test was used for analysing the correlation between levels of each gene with gestational age. After correlation values between genes were identified, a network graph was plotted using the qgraph R package [58] and illustrated using an adjacent matrix plot.

## Screening for regulators of PCOS candidate genes

For screening of possible regulators of PCOS candidate genes in the fetal ovary, RNA extracted from cultured and treated bovine fetal fibroblasts from another study was used [59]. All ovaries analysed in the current study were from the second trimester. Briefly, bovine fetal ovaries were collected, gestational ages of the fetuses were determined and all ovaries were transferred to the laboratory using the protocol previously mentioned. After removing the surrounded connective tissue, the ovaries were rinsed in 70% ethanol and HBSS$^{+/+}$, dissected and minced with a scalpel. The samples were digested in 1mg/ml collagenase type I (GIBCO/ Life Technologies Australia Pty Ltd, Mulgrave, VIC, Australia) in HBSS$^{+/+}$ at 37˚C shaking at 150 rpm and after centrifugation at 1500 rpm for 5 min, the supernatant was removed. The samples were then digested in 2 ml of 0.025% trypsin/EDTA (GIBCO/Life Technologies) in Hank's Balanced-Salt Solution without Mg$^{2+}$ and Ca$^{2+}$ (HBSS$^{-/-}$; Sigma-Aldrich) for 5 min at 37˚C at 150 rpm and centrifuged at 1500 rpm for 5 min. The pellet was then resuspended in DMEM/F12 medium containing 5% FCS, 1% penicillin and streptomycin sulphate, and 0.1% fungizone (all GIBCO/ Life Technologies). The cells were then dispersed and cultured in 6-well plates or 10 cm petri dishes at 38.5˚C and 5% CO$_2$ until confluent. The fetal fibroblast cultures were detached by treatment with 0.25% trypsin/EDTA, then the total number of viable cells was estimated with

the trypan blue method and stored in liquid nitrogen for subsequent experiments. Bovine fetal fibroblasts (n = 4 from weeks 13, 14, 17 and 19 of gestation for screening of possible regulators; n = 6 for weeks 19–26 of gestation for TGFβ-1 treatment) previously stored in liquid nitrogen were thawed and 30,000 cells/well seeded in 24-well plates in DMEM/F12 medium containing 5% FCS, 1% penicillin and streptomycin sulphate and 0.1% fungizone. The cells were incubated for 24 h at 38.5˚C and 5% $CO_2$ until 60–70% confluent then the wells were washed with 1X PBS. Different chemical treatments at concentrations previously reported in the literature [S2 Table [52]] including 5 ng/µl or 20 ng/µl TGFβ-1 in DMEM/F12 medium containing 1% FCS, 1% penicillin and streptomycin sulphate, and 0.1% fungizone were added then after 18 h, the cells were harvested for RNA.

## Results

### Expression of PCOS candidate genes in the human fetal ovary

We first analysed the mRNA expression levels of *FBN3*, *HMGA2*, *TOX3*, *GATA4*, *DENND1A. V1-7*, *DENND1A.V1,3,4*, *FSHB*, *LHCGR* and *FSHR* in morphologically normal human fetal ovaries less than 150 days of gestation (Fig 1). All genes, except for *FSHB*, were detected in the human fetal ovary (Fig 1). *FBN3*, *HMGA2* and *DENND1A.V1,3,4* were highly expressed before 70 days of gestation (Fig 1A, 1C and 1G), then their expression levels markedly decreased. Additionally, their expression levels were strongly correlated to each other (Table 1). The expression level of *GATA4* was strongly correlated with expression levels of *FBN3* ($P < 0.001$), *HMGA2* ($P < 0.0001$) and *DENND1A* ($P < 0.0001$) (Table 1). A weaker but significant correlation was observed between the *GATA4*, *DENND1A.V1-7* and *FSHR* expression levels and gestational age (Fig 1B, 1F and 1H and Table 1).

### Expression of PCOS candidate genes in bovine fetal and adult ovaries

The mRNA levels of PCOS candidate genes were determined in whole ovaries (n = 27) collected from bovine fetuses ranging from 8–39 weeks gestation and in adult ovaries and all the PCR products were sequenced to confirm their correct amplification (S1 Table). Most of the PCOS candidate genes were expressed during the development of the bovine fetus (Figs 2–4). We could not detect expression of *DENND1A.V2* and *SUMO1P1* in the bovine fetal ovary, nor in the bovine adult thecal cells, ovary, heart, spleen, kidney or liver.

Of the PCOS candidate genes that were analysed in the bovine fetal ovary, *FBN3*, *GATA4*, *HMGA2*, *TOX3* and *DENND1A.X1,2,3,4* were expressed early during ovarian development and then declined (Fig 2A–2D and 2G). *LHCGR* was upregulated in early development, downregulated to the lowest point around 150 days of gestation, then sharply increased in late gestation (Fig 2E), whereas the other early genes showed no similar increase in late gestation. *FSHB* mRNA was originally measured with primers located within exon 3 and these levels peaked at around 100 days of gestation then declined until the end of gestation, however, the levels were exceedingly low and may not have much biological relevance (Fig 2F). To examine *FSHB* further we designed primers to exons 2 and 3, and whilst they correctly amplified *FSHB* from the anterior pituitary they failed to amplify from the ovary samples, suggesting that there is transcriptional activity of *FSHB* in the fetal ovary, but the transcript cannot encode and produce FSHB. The mRNA levels of three of the PCOS candidate genes identified by GWAS studies, *INSR*, *FSHR* and *AMH*, as well as *AR* and *TGFB1I1* were low early in gestation gradually increasing until the end of gestation (Fig 3A–3E). The other genes, *C8H9orf3*, *RAB5B*, *ERBB4*, *YAP1*, *SUOX*, *RAD50*, *THADA*, and *KRR1* were expressed throughout development, however, their mRNA levels did not correlate with gestational age (Fig 4A–4H).

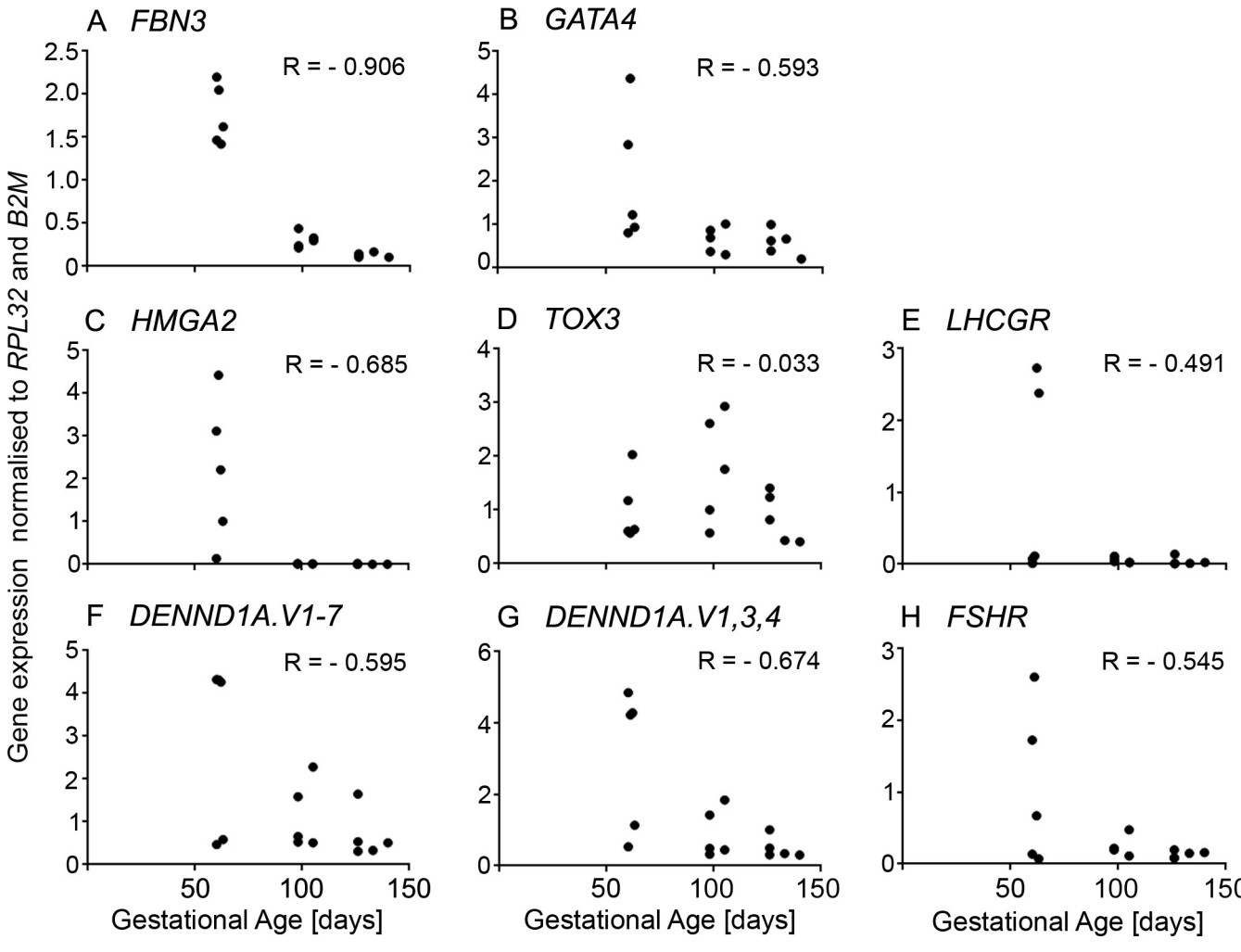

**Fig 1. Scatter plot of mRNA expression levels of some PCOS candidate genes analysed in human fetal ovaries (n = 15, 8–20 weeks of gestation).** Pearson correlation coefficient (r) test was used to analyse the data. *P* values are shown in Table 1.

**Table 1. Pearson correlation coefficients (r) of mRNA expression levels of PCOS-candidate genes in human fetal ovaries and their gestational ages (n = 15).**

|  | Age | FBN3 | GATA4 | HMGA2 | TOX3 | LHCGR | DENND1A.V1-7 | DENND1A.V1,3,4 |
|---|---|---|---|---|---|---|---|---|
| FBN3 | -0.906[b] |  |  |  |  |  |  |  |
| GATA4 | -0.593[a] | 0.767[c] |  |  |  |  |  |  |
| HMGA2 | -0.685[b] | 0.852[d] | 0.934[d] |  |  |  |  |  |
| TOX3 | -0.033 | -0.134 | -0.078 | -0.115 |  |  |  |  |
| LHCGR | -0.491 | 0.421 | 0.020 | 0.287 | 0.081 |  |  |  |
| DENND1A.V1-7 | -0.595[a] | 0.697[b] | 0.782[c] | 0.864[d] | 0.302 | 0.291 |  |  |
| DENND1A.V1,3,4 | -0.674[b] | 0.790[c] | 0.808[c] | 0.904[d] | 0.232 | 0.357 | 0.982[d] |  |
| FSHR | -0.545[a] | 0.575[a] | 0.680[b] | 0.606[a] | -0.185 | -0.033 | 0.445 | 0.421 |

[a] $P < 0.05$

[b] $P < 0.01$

[c] $P < 0.001$

[d] $P < 0.0001$; Pearson correlation tests.

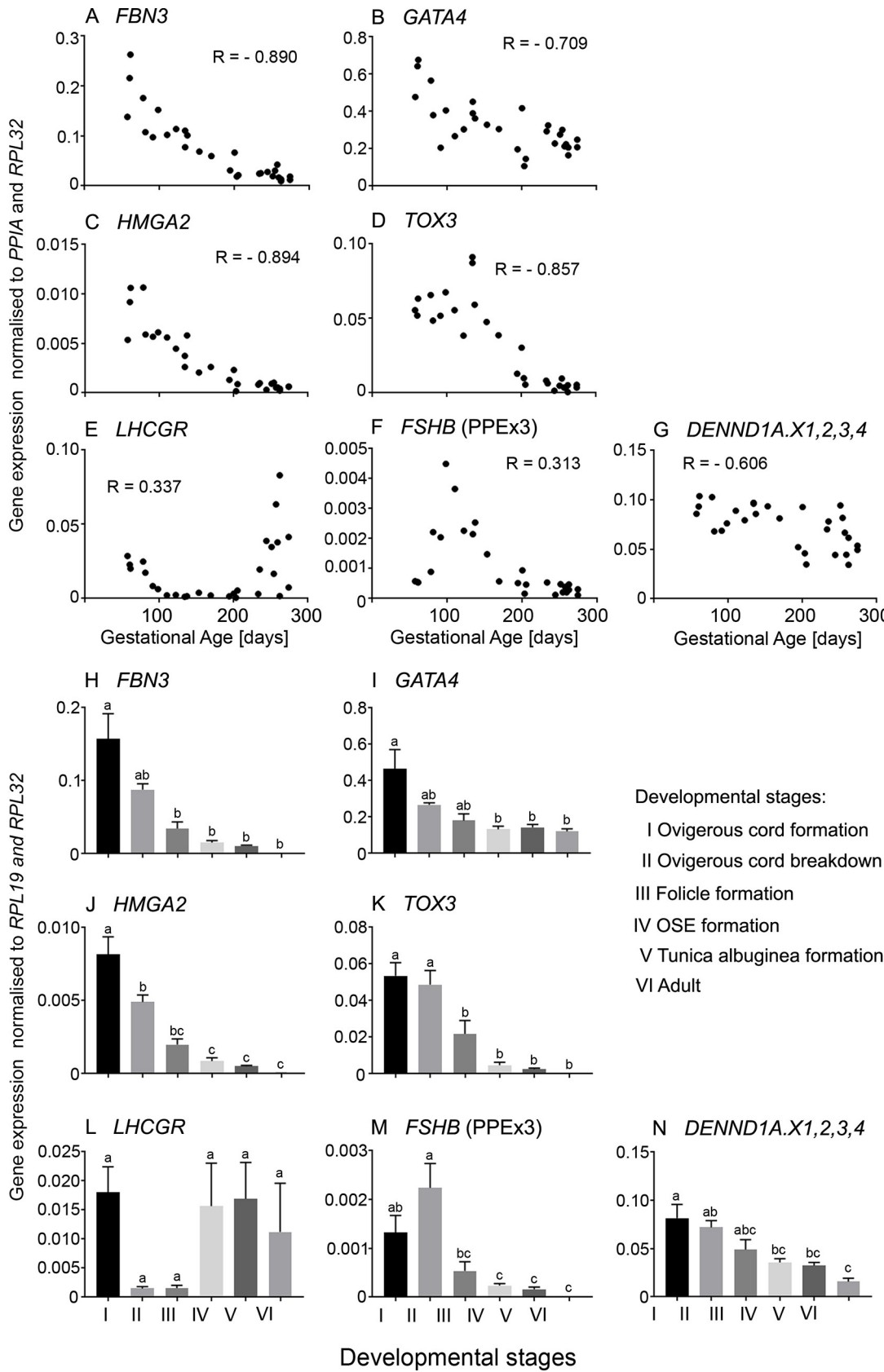

**Fig 2. A-G. Scatter plots of mRNA expression levels of PCOS candidate genes which were highly expressed during early gestation in bovine fetal ovaries (n = 27).** Only the primer pair in exon 3 for *FSHB* amplified, whereas primers from exons 2 and 3 did not (not shown), suggesting the *FSHB* transcript, whilst expressed cannot encode the full length FSHB in the ovary, as expected. Pearson correlation coefficient (R) test was used to analyse data and *P* values are shown in Table 2. **H-N.** Differential mRNA expression levels in ovaries grouped into six stages of ovarian development based on their histological morphology: ovigerous cord formation (n = 7, Stage I), ovigerous cord breakdown (n = 4, Stage II), follicle formation (n = 3, Stage III), surface epithelium formation (n = 8, Stage IV), tunica albuginea formation (n = 5, Stage V) and adult (n = 6, Stage VI). Data are presented as mean ± s.e.m. (normalised to *PPIA* and *RPL32* for scatter plot and *RPL19* and *RPL32* for bar graphs). One-way ANOVA with *post hoc* Holm-Sidak test were used to analyse the data. Bars with different letters are statistically significantly different from each other (P < 0.05).

We also classified the fetal ovary samples into 5 groups reflecting key stages in development as shown previously [37] and described in the Materials and Methods. We compared the mRNA levels at each stage and with those in the adult ovary. *FBN3*, *HMGA2* and *TOX3* mRNA levels were high in the early stages and decreased significantly from stage III to the lowest levels in the adult ovary (Fig 2H and 2J–2K). *GATA4*, *FSHB* and *DENND1A.X1,2,3,4* (Fig 2I, 2M and 2N), as well as *C8H9orf3*, *RAB5B*, *RAD50* and *KRR1* (Fig 4I, 4J, 4N and 4P), mRNA levels were elevated in stages I and II declining significantly at the later stages. There was no significant difference between the mRNA levels of *YAP1* and *SUOX* in all 5 fetal stages, however, in the adult ovary, their levels were significantly decreased (Fig 4L and 4M). *FSHR* and *AMH* were expressed highly in the adult ovary compared to all 5 stages in the fetal ovary (Fig 3G and 3H). The mRNA levels of *AR* and *TGFB1I1* peaked at stage V (Fig 3I and 3J). Their mRNA levels were significantly lower in the adult ovary compared to stages IV and V (Fig 3I and 3J). No significant differences were observed in the mRNA levels of *LHCGR* (Fig 2L), *INSR* (Fig 3F) and *THADA* (Fig 4O) in the fetal and adult ovaries.

## Correlations between levels of mRNA

To analyse the correlation between the expression levels of PCOS candidate genes and gestational age in bovine ovaries, we generated a Pearson correlation matrix (Table 2). *FBN3* positively correlated with *HMGA2* (P < 0.0001), *TOX3* (P < 0.0001), *GATA4* (P < 0.0001) and *DENND1A.X1,2,3,4* (P < 0.0001), *HMGA2* positively correlated with *TOX3* (P < 0.0001), *GATA4* (P < 0.0001) and *DENND1A.X1,2,3,4* (P < 0.001), *TOX3* positively correlated with *GATA4* (P < 0.0001) and *DENND1A.X1,2,3,4* (P < 0.0001), and *GATA4* positively correlated with *DENND1A.X1,2,3,4* (P < 0.0001) (Table 2). In addition, *FBN3*, *HMGA2* and *TOX3* also negatively correlated (r > -0.6) with *FSHR*, *AMH*, *INSR*, *AR* and *TGFB1I1*. After correlation values between genes were identified, a network graph was plotted using the qgraph R package [58] to plot an adjacent matrix [S5 Fig [52]]. Most of the genes, except *ERBB4*, were closely and highly connected with each other as well as with gestational age, suggesting a strong correlation between all genes and gestational age.

To accommodate the bimodal expression pattern of *LHCGR*, we additionally analysed the correlations between expression levels of genes in bovine fetal ovaries using the data from fetuses less than 150 days of gestation [S3 Table [52]]. Our results showed that there was a strong correlation (r > 0.6) between the expression levels of the majority of genes that were highly expressed in early development (*FBN3*, *GATA4*, *HMGA2*, *DENND1A.X1,2,3,4* and *LHCGR*) [S3 Table [52]]. Interestingly, a strong negative correlation was observed between *LHCGR* and *INSR* (P < 0.01), *AR* (P < 0.01) and *TGFB1I1* (P < 0.001), which are the genes that were highly expressed in late stages of development [S3 Table [52]]. The results also showed a strong correlation (r > 0.6) between *FSHB* and *GATA4* and between *HMGA2* and *LHCGR* [S3 Table [52]].

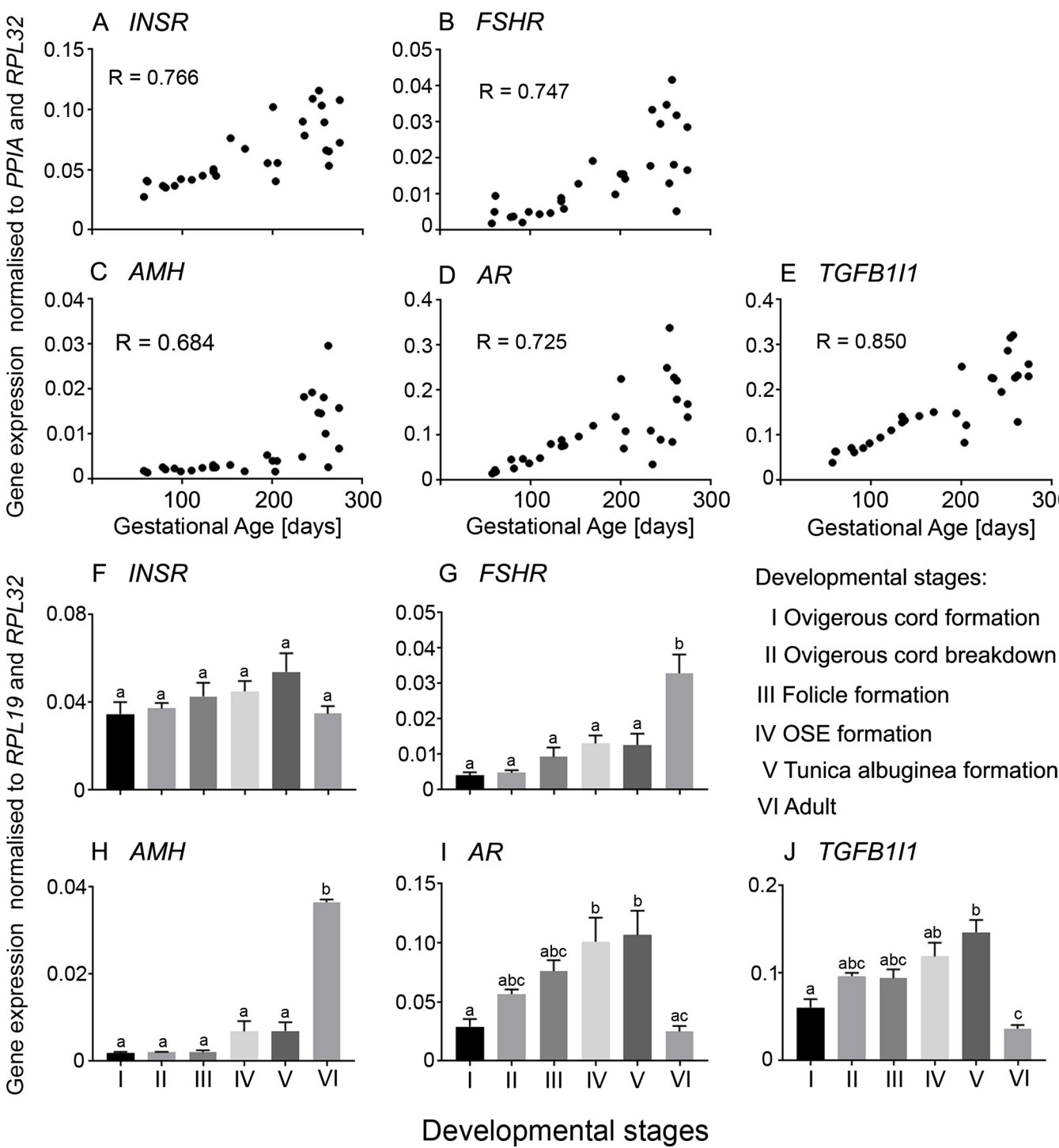

**Fig 3. A-E. Scatter plots of mRNA expression levels from genes which are highly expressed late in gestation in bovine fetal ovaries (n = 27).** Pearson correlation coefficient (R) test was used to analyse data and *P* values are shown in Table 2. **F-J**. Differential mRNA expression levels in ovaries grouped into six stages of ovarian development based on their histological morphology: ovigerous cord formation (n = 7, Stage I), ovigerous cord breakdown (n = 4, Stage II), follicle formation (n = 3, Stage III), surface epithelium formation (n = 8, Stage IV), tunica albuginea formation (n = 5, Stage V) and adult (n = 6, Stage VI). Data are presented as mean ± s.e.m. (normalised to *PPIA* and *RPL32* for scatter plot and *RPL19* and *RPL32* for bar graphs). One-way ANOVA with *post hoc* Holm-Sidak test were used to analyse the data. Bars with different letters are statistically significantly different from each other (P < 0.05).

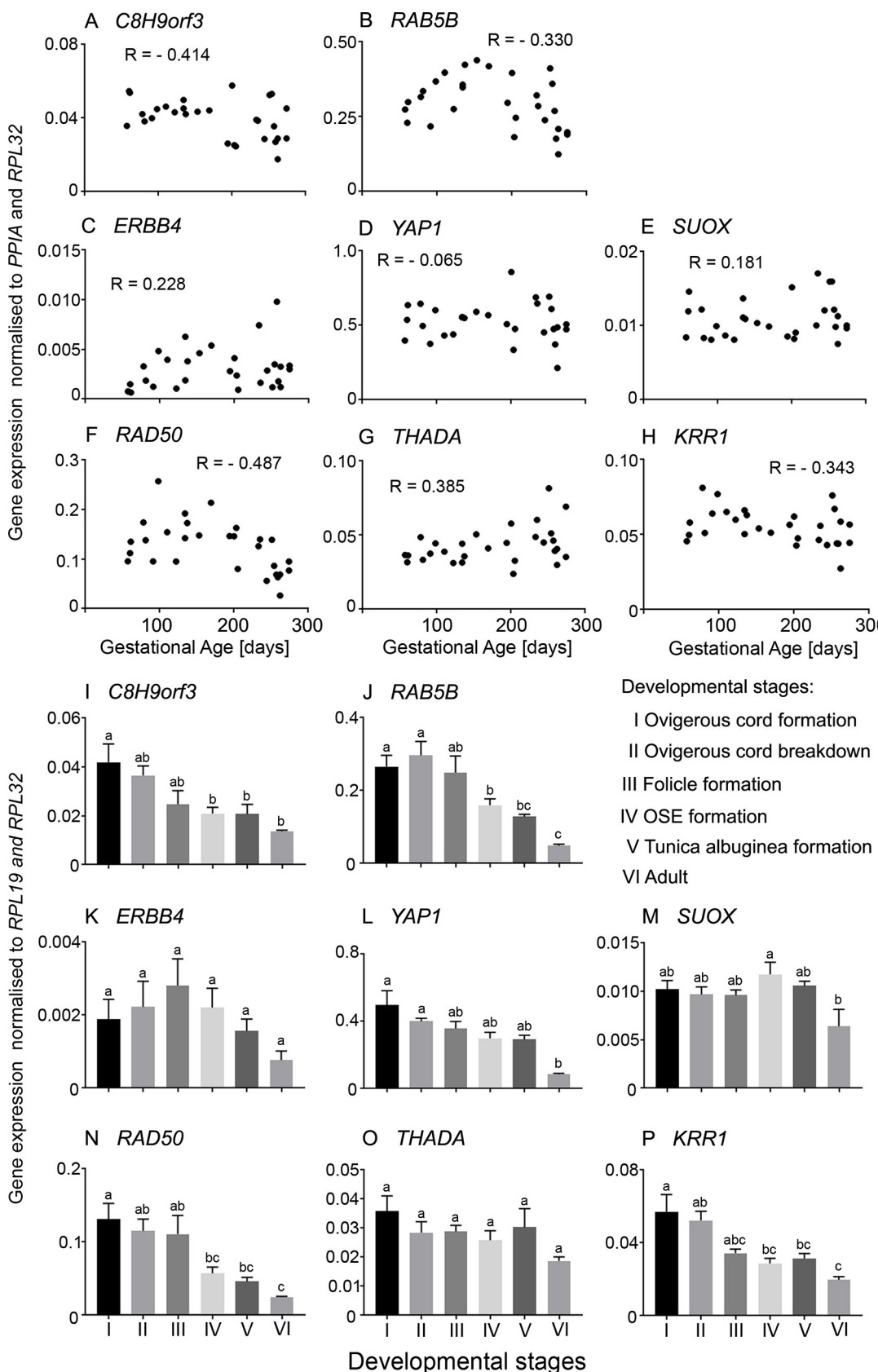

**Fig 4. A-H. Scatter plot of mRNA expression levels of PCOS candidate genes which are constantly expressed throughout gestation in bovine fetal ovaries (n = 27).** Pearson correlation coefficient (R) test was used to analyse data and *P* values are shown in Table 2. **I-M.** Differential mRNA expression levels in ovaries grouped into six stages of ovarian development based on their histological morphology: ovigerous cord formation (n = 7, Stage I), ovigerous cord breakdown (n = 4, Stage II), follicle formation (n = 3, Stage III), surface epithelium formation (n = 8, Stage IV), tunica albuginea formation (n = 5, Stage V) and adult (n = 6, Stage VI). Data are presented as mean ± s.e.m. (normalised to *PPIA* and *RPL32* for scatter plot and *RPL19* and *RPL32* for bar graphs). One-way ANOVA with *post hoc* Holm-Sidak test were used to analyse the data. Bars with different letters are statistically significantly different from each other (P < 0.05).

## Differential gene expression in the cortex and medulla of fetal ovary

To study the localisation of cells expressing these genes, we conducted laser capture micro-dissection in the bovine ovarian cortex and medulla at early (10–17 weeks) and late (36–39 weeks) stages of gestation. We investigated the mRNA levels of PCOS candidate genes in these two areas [S6–S8 Figs [52]]. If expressed higher in the medulla we putatively interpreted this as being stromal expression, as this area is devoid of GREL cells, granulosa cells and germ cells, and if expressed more highly in the cortex, rich in GREL cells, granulosa cells and germ cells, we putatively interpreted this as expression in the cortical ovigerous cords or follicles if late in gestation. Validation of this approach was confirmed by examining the expression genes previously immunolocalised in bovine fetal ovaries [35,40]. These included markers of germ cell (*VASA* and *OCT4*) as well as the epithelial marker *KRT19*, which is expressed in GREL and granulosa cells of the bovine fetal ovary but not in stromal cells [35] [S9 Fig [52]], and *FBN3* [S6 Fig [52]], which is expressed in the stroma [35,40].

The level of mRNA of *FSHB* was significantly higher in the ovarian cortex compared with the medulla in the early stage [S6 Fig [52]], *FSHR* in the late stage [S7 Fig [52]] and *ERBB4* in both the early and late stage of development [S8 Fig [52]]. The expression levels of *FBN3* [S6 Fig [52]], *INSR*, *AR*, *TGFB1I1* [S7 Fig [52]], *YAP1* and *SUOX* [S8 Fig [52]] were significantly higher in the ovarian medulla compared with the cortex in early development. Other genes did not show any significant difference between cortical and medullar expression levels.

## Regulation of PCOS related genes in the bovine fetal ovary

We treated bovine fetal fibroblasts (n = 4 from 13–19 weeks of gestation) with 24 growth factors and hormones [S2 Table [52]], and observed the effect on the expression of PCOS candidate genes [S10–S12 Figs [52]]. These factors and hormones have been shown to have important physiological roles in the adult ovary, such as cell proliferation and the production of extracellular matrices. The average levels of expression ($2^{-\Delta\Delta-Ct}$) for the untreated cells are shown in S4 Table [52], and compared with the *in vivo* RNA levels from whole ovaries from the same gestational ages as shown in Figs 2–4. Most genes were expressed at a lower level except *GATA4* and *TGFB1I1* which were >20 fold higher *in vitro* [S4 Table [52]]. Treatment with fibroblast growth factor 9 (FGF-9) significantly increased mRNA levels of *HMGA2* in fetal fibroblasts [S10 Fig [52]]. Other treatments did not affect the expression level of *HMGA2* [S10–S12 Figs [52]]. Other PCOS genes, except *FSHR* and *FSHB*, were detected in the cultured bovine fetal fibroblasts, however, none were significantly affected by any of the treatments [S10–S12 Figs [52]].

Treatment with TGFβ-1 (5 or 20 ng/ml) of 19–26 weeks bovine fetal fibroblasts resulted in a significant reduction in expression levels of *INSR*, *AR*, *C8H9orf3* and *RAD50* (Fig 5), compared to the untreated control and this reduction was enhanced by the higher TGFβ-1 concentration. We did not find any significant effect on *TGFB1I1* expression in fetal fibroblasts treated with 5 ng/ml TGFβ-1, however, a significant increase was observed with 20 ng/ml TGFβ-1 (Fig 5). Other genes were not significantly affected (Fig 5).

**Table 2. Pearson correlation coefficients (r) of mRNA expression levels of PCOS-candidate genes and gestational age in bovine fetal ovaries (n = 27).**

| | Age | FBN3 | GATA4 | HMGA2 | TOX3 | LHCGR | FSHB | DENND1A. X1,2,3,4 | C8H9orf3 | RAB5B | ERBB4 | YAP1 | SUOX | RAD50 | THADA | KRR1 | INSR | FSHR | AMH | AR |
|---|---|---|---|---|---|---|---|---|---|---|---|---|---|---|---|---|---|---|---|---|
| FBN3 | -0.887[d] | | | | | | | | | | | | | | | | | | | |
| GATA4 | -0.709[d] | 0.898[d] | | | | | | | | | | | | | | | | | | |
| HMGA2 | -0.898[d] | 0.959[d] | 0.849[d] | | | | | | | | | | | | | | | | | |
| TOX3 | -0.892[d] | 0.824[d] | 0.670[d] | 0.826[d] | | | | | | | | | | | | | | | | |
| LHCGR | 0.332 | -0.156 | -0.079 | -0.177 | -0.382[a] | | | | | | | | | | | | | | | |
| FSHB (PPEx3)* | -0.563[b] | 0.406[a] | 0.182 | 0.443[a] | 0.678[c] | -0.403[a] | | | | | | | | | | | | | | |
| DENND1A. X1,2,3,4 | -0.606[c] | 0.696[d] | 0.777[d] | 0.675[c] | 0.731[d] | -0.311 | 0.347 | | | | | | | | | | | | | |
| C8H9orf3 | -0.393[a] | 0.543[b] | 0.671[c] | 0.506[b] | 0.511[b] | -0.256 | 0.295 | 0.841[d] | | | | | | | | | | | | |
| RAB5B | -0.314 | 0.246 | 0.340 | 0.258 | 0.485[a] | -0.459[a] | 0.471[a] | 0.714[d] | 0.644[c] | | | | | | | | | | | |
| ERBB4 | 0.305 | -0.240 | -0.161 | -0.256 | -0.164 | 0.162 | 0.049 | 0.010 | 0.044 | 0.278 | | | | | | | | | | |
| YAP1 | -0.080 | 0.236 | 0.491[b] | 0.217 | 0.225 | -0.387[a] | 0.064 | 0.637[c] | 0.716[d] | 0.692[d] | 0.232 | | | | | | | | | |
| SUOX | 0.175 | 0.140 | 0.403[a] | 0.083 | -0.037 | 0.073 | -0.250 | 0.505[b] | 0.639[c] | 0.392[a] | 0.073 | 0.722[d] | | | | | | | | |
| RAD50 | -0.497[b] | 0.405[a] | 0.359 | 0.434[a] | 0.600[c] | -0.577[b] | 0.563[b] | 0.526[b] | 0.430[a] | 0.690[d] | 0.186 | 0.490[b] | 0.054 | | | | | | | |
| THADA | 0.380 | -0.264 | 0.001 | -0.231 | -0.268 | 0.177 | -0.176 | 0.217 | 0.435[a] | 0.359 | 0.225 | 0.557[b] | 0.585[b] | 0.059 | | | | | | |
| KRR1 | -0.352 | 0.379 | 0.368 | 0.463[a] | 0.515[b] | -0.478[a] | 0.505[b] | 0.663[c] | 0.586[b] | 0.600[c] | -0.110 | 0.589[b] | 0.430[a] | 0.591[b] | 0.334 | | | | | |
| INSR | 0.775[d] | -0.601[c] | -0.312 | -0.643[c] | -0.632[c] | 0.248 | -0.435[a] | -0.143 | 0.167 | 0.116 | 0.369 | 0.392[a] | 0.572[b] | -0.283 | 0.756[d] | -0.070 | | | | |
| FSHR | 0.753[d] | -0.580[b] | -0.436[d] | -0.640[c] | -0.684[d] | 0.675[c] | -0.531[b] | -0.392[a] | -0.203 | -0.196 | 0.451[a] | -0.062 | 0.256 | -0.391[a] | 0.456[a] | -0.432[a] | 0.729[d] | | | |
| AMH | 0.682[d] | -0.525[b] | -0.408[a] | -0.542[b] | -0.632[c] | 0.846[d] | -0.427[a] | -0.476[a] | -0.310 | -0.374 | 0.242 | -0.306 | 0.153 | -0.619[c] | 0.313 | -0.438[a] | 0.573[b] | 0.818[d] | | |
| AR | 0.765[d] | -0.669[c] | -0.420[a] | -0.654[c] | -0.648[c] | 0.234 | -0.452[a] | -0.258 | -0.028 | -0.047 | 0.055 | 0.137 | 0.420[a] | -0.358 | 0.491[b] | -0.040 | 0.696[d] | 0.476[a] | 0.568[b] | |
| TGFB1I1 | 0.853[d] | -0.664[c] | -0.413[a] | -0.695[c] | -0.676[c] | 0.231 | -0.444[a] | -0.188 | 0.034 | 0.024 | 0.432[a] | 0.292 | 0.518[b] | -0.352 | 0.584[b] | -0.032 | 0.876[d] | 0.669[c] | 0.537[b] | 0.772[d] |

[a] $P < 0.05$

[b] $P < 0.01$

[c] $P < 0.001$

[d] $P < 0.0001$; Pearson correlation tests.

*Primer pair from exon 3.

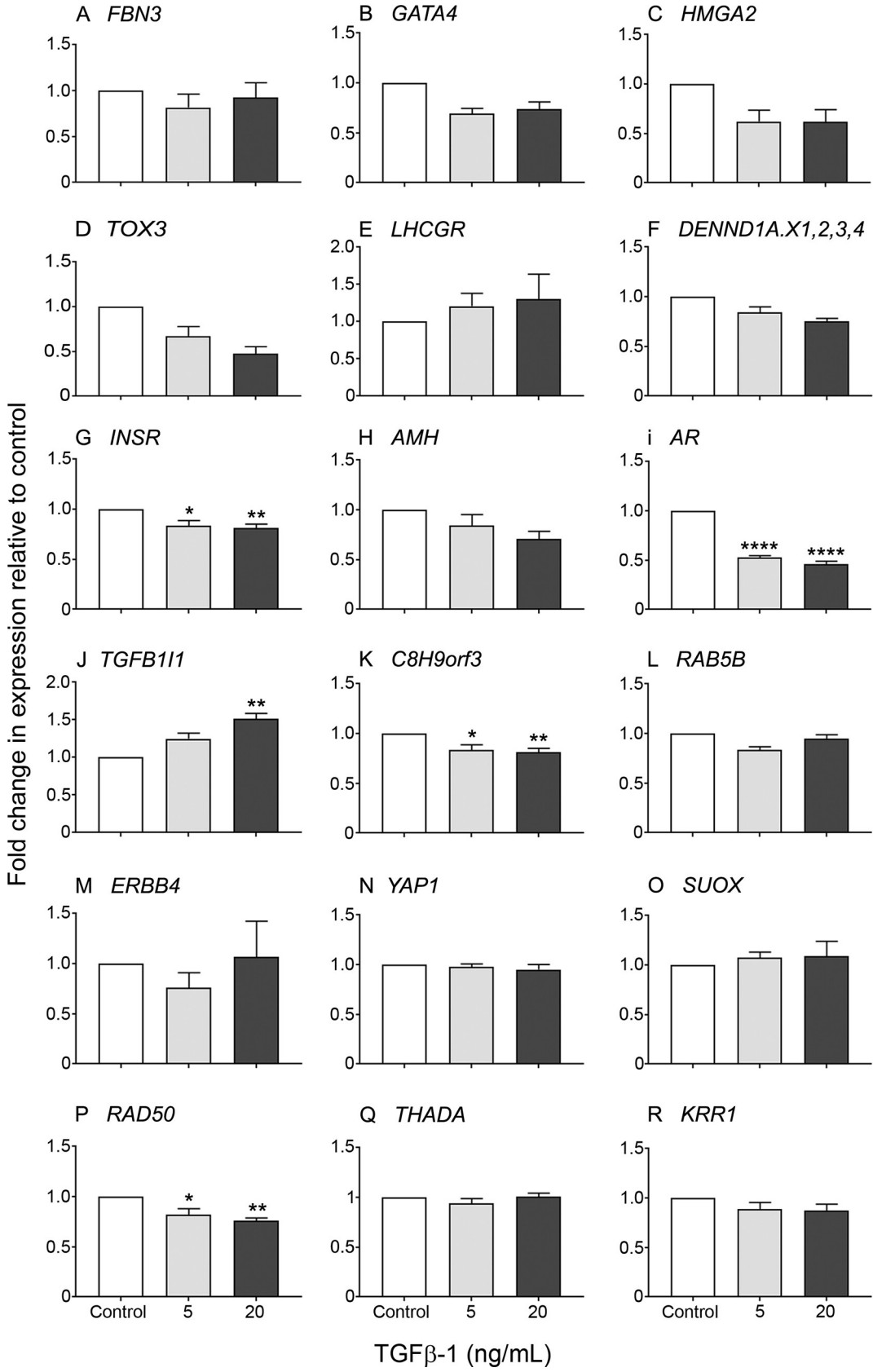

**Fig 5. Expression of genes in bovine fetal fibroblasts from 19–26 weeks of gestation.** Fibroblasts were cultured in the presence of 5 and 20 ng/mL TGFβ-1 for 18 h. Data are represented as mean ± s.e.m. of fold change in gene expression relative to the untreated control (n = 6 ovaries). One-way ANOVA with Dunnet's *post hoc* test were used to analyse the data. Asterisk symbols indicates statistically significant different form the control. $^{*}P < 0.05$, $^{**}P < 0.01$.

## Discussion

This study reports the levels of mRNA of PCOS candidate genes identified by GWAS and other genes in the developing human and bovine fetal ovaries. We were able to identify a number of candidate genes that were expressed, whose pattern of expression changed during gestation, and whose expression levels were highly correlated with each other. Additionally, in bovine ovaries we identified the region of the ovary where the genes were expressed and we examined gene expression in cultured fetal ovarian fibroblasts. This study thus identified relationships between the genetic and anatomical development of the ovary in fetal life. It should be noted that this study was limited to the analyses of 15 human fetal ovaries from less than 150 days of gestation, which is why we additionally undertook the bovine analyses. No direct causality studies were conducted. It is difficult to imagine how they could be at this stage, given that human ovaries cannot be sampled during fetal development and linked with PCOS status in later life.

For ease of discussion we have grouped the genes into three categories based upon their expression patterns across gestation. The early genes were expressed highly in the first trimester but continued to decline in expression to reach a nadir at about mid gestation. The late genes were first detected midway through gestation and continued to increase in expression during the remainder of gestation. The third group of genes was expressed at relatively similar levels across all of gestation. In bovine samples expression of some genes including *DENND1A.V2* and *SUMO1P1* (bovine only examined) was not detected. A transcript from exon 3 of *FSHB* was detected in early bovine ovaries but this could not encode the full length FSHB as no transcript with primers spanning an intron from exons 2 and 3 was detected.

Earlier studies of a selection of 44 genes associated with stromal and thecal cells across gestation in the bovine [45] or with germ, GREL and granulosa cells [60] have been undertaken. In the former study expression levels of *FBN3*, *LHCGR*, *AR* and *TGFB1I1* were also measured as in the current study. The tissues used in that study are not the same as in the current study and are not as young as some of the current ones. The expression pattern of these four genes was confirmed in the current study giving confidence to the reproducibility of the results. Previously we performed hierarchical clustering analysis of the 44 genes and they also segregated into three clusters expressed early in the first trimester or late in the third trimester or with their expression increasing in the second and remaining high in the third trimester [45].

### The early genes

The group of bovine genes expressed early in gestation included *FBN3*, *GATA4*, *HMGA2*, *TOX3* and *DENND1A.X1,2,3,4*. Their mRNA levels were strongly positively correlated with each other and negatively with gestational age. *FBN3*, *GATA4* and *DENND1A.X1,2,3,4* were elevated in the medulla early in gestation but *HMGA2* and *TOX3* did not show any differential pattern between cortex and medulla. *LHCGR* expression was also elevated early during ovarian development declining to the lowest point around mid-gestation, before sharply increasing until the end of gestation. *LHCGR* expression was also elevated in the medulla at the early stages, like *FBN3*, *GATA4* and *DENND1A.X1,2,3,4*. Examining the data in the first half of gestation only, showed that *LHCGR* expression was positively but weakly correlated with most of the other early genes. This early expression of *LHCGR* precedes follicle formation [35] and the

expression in the medulla suggests that this is not associated with follicular cells at the early stage.

This same set of genes was examined in human fetal ovaries in the first half of gestation. All genes identified in the bovine were expressed in the human ovaries, particularly at the earliest stages. In the human ovary *FBN3*, *GATA4*, *HMGA2* and *DENND1A* (V1-7 and V1, 3, 4) were all strongly positively correlated with each other and negatively with gestational age. *TOX3* only declined in the two oldest samples examined and *LHCGR* was only elevated in two of four youngest samples. Thus whilst these genes showed similar patterns of expression to the other early genes their expression was not correlated with them.

Two other genes were expressed early but differently between bovine and human. A transcript from exon 3 of *FSHB*, although normally only expressed in the anterior pituitary, was detected early in the bovine ovary and its expression level peaked at around 100 days of gestation then declined until the end of gestation. However, it should be noted that the levels were very low and unlikely to encode FSHB as no transcript was detected using primers from exons 2 and 3. Therefore the FSHB expression may not be of biological relevance but its differential expression is of potential interest. In the human, *FSHB* was not detected but *FSHR* was expressed in three of the youngest four ovaries examined and it was very low in the older samples. At this stage it is not entirely clear how best to interpret these findings.

*DENND1A* encodes DENND1A (DENN domain containing 1A, connecdenn 1) protein which is believed to assist endocytosis [61]. Human *DENND1A* possesses 7 variants in total (*DENND1A.V1-7*) as well as 19 predicted variants (DENND1A.X1-19) (https://www.ncbi.nlm.nih.gov/). The function of these variants is still unclear, however, 2 variants, *DENND1A.V1* and *DENND1A.V2* have been linked to PCOS [62,63]. *DENND1A.V1* encodes a 1009 amino acid protein with C-terminal proline-rich domain, whereas *DENND1A.V2* encodes a 559 amino acid protein containing three DENN domains and additionally a C-terminal 33 amino acid sequence [62]. A recent study has found that *DENND1A.V2* might play an important role in pathophysiology of hyperandrogenaemia associated with PCOS [62] due to several reasons: significantly elevated levels of *DENND1A.V2* mRNA and protein in PCOS thecal cells compared to non-PCOS thecal cells; significantly elevated levels of *DENND1A.V2* mRNA in urine of PCOS women compared to women with normal cycles; augmentation of *CYP17A1* and *CYP11A1* gene transcription, as well as increased dehydroepiandrosterone production in response to overexpression of *DENND1A.V2* in normal thecal cells [63]. We were unable to detect the expression of *DENND1A.V2* in bovine fetal ovaries or adult thecal cells. However, there were elevated mRNA levels of *DENND1A* in bovine and human fetal ovaries early in development.

Overall this group of genes expressed early has some unique characteristics. The genes are highly expressed when the ovigerous cords are forming by penetration of the stroma from the mesonephros [35,37] and the proteins encoded by these genes have different cellular functions including extracellular matrix, transcription factors, hormone receptors and clathrin-mediated endocytosis. These genes may also be expressed in different cell types as inferred from their different expression patterns comparing the cortex and medulla of the ovary. These genes are all elevated long before the time of gestation when animal models of PCOS can be generated by treating pregnant mothers or newborns with androgens [64].

## The late genes

The mRNA levels of two of the PCOS candidate genes identified by GWAS, *INSR* and *FSHR*, as well as *AMH*, *AR* and *TGFB1I1* were low early in bovine gestation and then gradually increased from mid gestation until the end of gestation. The mRNA levels of these genes were

highly correlated with each other and with gestational age. *INSR*, *AR* and *TGFB1I1* were elevated in the medulla suggesting that they are expressed in the stroma. *FSHR* was elevated in the cortex in late gestation suggesting that it is associated, as expected, with granulosa cells of follicles. The *LHCGR* gene was also upregulated in the second half of gestation following a nadir in mid gestation, and its mRNA levels were highly correlated with *FSHR* and *AMH*, but not *INSR*, *AR* and *TGFB1I1* mRNA levels. It is intriguing that expression levels correlate with each other yet three of the genes appear to be expressed in stroma (*INSR*, *AR* and *TGFB1I1*) and three in follicles (*FSHR*, *LHCGR* and *AMH*). Expression of *TGFB1I1* and *AR* suggest that TGFβ signalling could be augmenting AR signalling and regulating stromal cell function during the second half of gestation.

## The constantly expressed genes

The other genes, *C8H9orf3*, *RAB5B*, *ERBB4*, *YAP1*, *SUOX*, *RAD50*, *THADA* and *KRR1* were expressed throughout development of the bovine fetal ovary, however, their mRNA levels did not correlate with gestational age except for weak negative correlations with *C8H9orf3* and *RAD50*. Within this group there were a number of associations, but in particular *C8H9orf3*, *RAB5B*, *SUOX* and *YAP1* were highly positively correlated with each other and with *DENND1A.X1,2,3,4* from the early group of genes. *C8H9orf3* also showed some weaker but positive correlations with some of the other members of the early group of genes. *SUOX* and *YAP1* were elevated in the medulla and *ERBB4* was elevated in the cortex. There is little information regarding the expression and function of *SUMO1P1*, one of the eight human *SUMO1* pseudogenes in human or bovine [65]. In this study, we could not detect expression of *SUMO1P1* in the bovine fetal ovary or adult thecal cells, liver, heart, spleen and kidney, suggesting that this gene might not be expressed in bovine tissues.

## Overall changes in PCOS candidate gene expression

The mRNA levels of most PCOS candidate genes were correlated with a number of others in bovine fetal ovaries, suggesting a degree of co-regulation or coordination of behaviours of different cells in the ovary. Genes that were highly expressed early could be linked to stroma development, whereas genes that were upregulated later in the development may have an important role in follicular development. Interestingly, strong negative correlations were observed between the genes that were upregulated early and those that were upregulated later in ovarian development.

## Regulation of genes

Bovine fetal fibroblasts from the second trimester were treated with growth factors and hormones and the expression of PCOS candidate genes measured. Previous examination of these cells in culture found that *FBN3* expression declined whilst *FBN1* increased [66] with time in culture, suggesting that the fibroblasts were undergoing maturation similar to changes in fibrillin gene expression during fetal ovarian development [40]. Most genes were unresponsive to treatments except *HMGA2* which increased in response to fibroblast growth factor 9 (FGF-9), and *INSR*, *AR*, *C8H9orf3* and *RAD50* all decreased while *TGFB1I1* increased in response to TGFβ-1. All the components of the TGFβ pathways are expressed in the fetal ovary, changing during the course of ovarian development [40]. Since TGFβ stimulates stromal growth and collagen deposition [42] the concept was developed that increased TGFβ bioactivity during fetal development could contribute the mechanism of the fetal disposition to PCOS in later life [40,43]. That five of the PCOS candidate genes, including *AR* and its co-activator *TGFB1I1*

could be regulated by TGFβ-1 further supports the concept that altered TGFβ signalling during fetal development could contribute to the aetiology of the predisposition to PCOS in later life.

## Summary and conclusions

In summary nearly all genes in loci associated with PCOS were expressed in the developing fetal ovary. They fell into three groups of early, late and constant expression during development. Within each group, the expression levels of many genes were strongly correlated with each other, despite, in some instances, being expressed in different areas of the ovary. This potentially indicates that gene expression is coordinated and linked to fetal ovarian development. Three of the PCOS candidate genes identified by GWAS, and *AR* and its coactivator *TGFB1I1*, were regulated by TGFβ *in vitro*. This is notable as fetal or neonatal exposure to androgens can induce PCOS in later life. Additionally in stroma TGFβ stimulates expansion of the stroma and collagen deposition, both known to be elevated in the polycystic ovary. Members of the TGFβ pathway also change during development of the ovary suggesting that TGFβ signalling is multifaceted [40]. With so many relationships between PCOS candidate genes during development of the fetal ovary, including TGFβ and androgen signalling, we suggest that future studies should determine if perturbations of these genes in the fetal ovary can lead to PCOS in later life.

## Supporting information

**S1 Fig. Representative images of the bovine fetal ovary showing the cortical and medullar areas prior to and post LCM.**
(PDF)

**S2 Fig. Alignment of the exons of human *DENND1A.V1*; bovine *DENND1A*; predicted bovine *DENND1A.X1,2,3,4*; and *DENND1A.X1,2,3,4* primer sequences.**
(PDF)

**S3 Fig. Alignment of the exons of human *DENND1A.V1*, *V3* and *V4* and *DENND1A.V1,3,4* primer sequences.**
(PDF)

**S4 Fig. Alignment of human *DENND1A.V2*; bovine *DENND1A*; and predicted bovine *DENND1A.V2* primer sequences.**
(PDF)

**S5 Fig. An adjacent matrix network graph of bovine gene expression using correlation coefficients from Table 2 generated with R-program.**
(PDF)

**S6 Fig. Differential mRNA expression levels of PCOS candidate genes which are highly expressed during early gestation in the cortex and medulla of bovine fetal ovaries.**
(PDF)

**S7 Fig. Differential mRNA expression levels of genes which are highly expressed late in gestation in the cortex and medulla of bovine fetal ovaries.**
(PDF)

**S8 Fig. Differential mRNA expression levels of PCOS candidate genes which are steadily expressed throughout gestation in the cortex and medulla of bovine fetal ovaries.**
(PDF)

**S9 Fig. Differential mRNA expression of control genes in the cortex and medulla of bovine fetal ovaries.**
(PDF)

**S10 Fig.** Expression of PCOS candidate genes which are highly expressed during early gestation in bovine fetal fibroblasts from less than 150 days of gestation cultured in the presence of treatments grouped as indicated (A to F) for 18 h.
(PDF)

**S11 Fig.** Expression of PCOS candidate genes which are highly expressed during late gestation in bovine fetal fibroblasts from less than 150 days of gestation cultured in the presence of treatments grouped as indicated (A to F) for 18 h.
(PDF)

**S12 Fig.** Expression of PCOS candidate genes which are highly expressed steadily during gestation in bovine fetal fibroblasts from less than 150 days of gestation cultured in the presence of treatments grouped as indicated (A to F) for 18 h.
(PDF)

**S1 Table. List of genes and primers used for qRT-PCR.**
(PDF)

**S2 Table. Treatments used for bovine fetal fibroblasts.**
(PDF)

**S3 Table. Pearson correlation coefficients (r) of mRNA expression levels of PCOS-candidate genes and gestational age in less than 150 day bovine fetal ovaries (n = 27).**
(PDF)

**S4 Table. Normalised gene expression in cultured bovine fetal fibroblast (in vitro, n = 4) and whole bovine fetal ovaries (in vivo, n = 5) at 13–28 weeks of gestation.**
(PDF)

## Acknowledgments

The authors would like to thank Thomas Food International, Murray Bridge, South Australia and Midfield Meat International, Warrnambool, Victoria for providing the bovine tissues for this research. We are also grateful to Ms Ruth Williams (Adelaide Microscopy, The University of Adelaide, North Terrace, Adelaide) for providing training on the Lasercapture Micro-dissection Microscope and to Adelaide Microscopy, The University of Adelaide, North Terrace, Adelaide for the use of the Leica AS-LMD Laser Microdissection Microscope.

## Author Contributions

**Conceptualization:** Katja Hummitzsch, Helen F. Irving-Rodgers, Richard A. Anderson, Raymond J. Rodgers.

**Investigation:** Monica D. Hartanti, Roseanne Rosario, Katja Hummitzsch, Nicole A. Bastian, Nicholas Hatzirodos, Wendy M. Bonner, Rosemary A. Bayne, Helen F. Irving-Rodgers.

**Project administration:** Raymond J. Rodgers.

**Supervision:** Katja Hummitzsch, Helen F. Irving-Rodgers, Richard A. Anderson, Raymond J. Rodgers.

**Writing – original draft:** Monica D. Hartanti.

**Writing – review & editing:** Roseanne Rosario, Katja Hummitzsch, Nicole A. Bastian, Nicholas Hatzirodos, Helen F. Irving-Rodgers, Richard A. Anderson, Raymond J. Rodgers.

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
