## [Decision Letter · Decision Letter 0]

9 Jan 2020

PONE-D-19-32933

Could perturbed fetal development of the ovary contribute to the development of polycystic ovary syndrome in women?

PLOS ONE

Dear Dr. Rodgers,

Thank you for submitting your manuscript to PLOS ONE. After careful consideration, we feel that it has merit but does not fully meet PLOS ONE’s publication criteria as it currently stands. Therefore, we invite you to submit a revised version of the manuscript that addresses the points raised during the review process.

The manuscript and the reviewers’ comments were carefully evaluated. The manuscript was appreciated by the Reviewers. Nevertheless, as suggested, the manuscript requires some improvement before to be considered for publication, particularly about the statistical methods, the discussion of results, and, particularly, the conclusion, that was highlighted by Reviewers as a point of important revision and possible misunderstanding of study aims. The suggested revisions are in detail reported in the Reviewers’ comments.

We would appreciate receiving your revised manuscript by Feb 23 2020 11:59PM. To enhance the reproducibility of your results, we recommend that if applicable you deposit your laboratory protocols in protocols.io, where a protocol can be assigned its own identifier (DOI) such that it can be cited independently in the future. For instructions see: http://journals.plos.org/plosone/s/submission-guidelines#loc-laboratory-protocols

We look forward to receiving your revised manuscript.

Kind regards,

Simone Garzon

Academic Editor

PLOS ONE

Journal Requirements:

Reviewers' comments:

Reviewer's Responses to Questions

**Comments to the Author**

1. Is the manuscript technically sound, and do the data support the conclusions?

Reviewer #1: Yes

Reviewer #2: Partly

Reviewer #3: Yes

Reviewer #4: Partly

2. Has the statistical analysis been performed appropriately and rigorously? 

Reviewer #1: No

Reviewer #2: Yes

Reviewer #3: Yes

Reviewer #4: Yes

3. Have the authors made all data underlying the findings in their manuscript fully available?

Reviewer #1: No

Reviewer #2: Yes

Reviewer #3: Yes

Reviewer #4: Yes

4. Is the manuscript presented in an intelligible fashion and written in standard English?

Reviewer #1: Yes

Reviewer #2: Yes

Reviewer #3: Yes

Reviewer #4: No

5. Review Comments to the Author

Reviewer #1: The manuscript is very interesting as it addresses the expression of many important markers of PCOS in the ovary in two models: human and bovine. Nevertheless, there are a few important concerns:

1) Clinical data about the mothers and cause of miscarriage is needed. Are there any cases with chromosomopathies involved?

2) Figure 1: is gestational age a continuos variable here or are the cases grouped ( they seem to represent 3 columns). If the graph represents 2 continuous variables please report de p value for each of the correlations shown. To state that in the human model there is a significant decrease or change in the markers either classify them in groups as with the bovine model and do a statistical comparison or generate a model. Only the R value is given which is not enough.

3)Figure 2 H to N please explain what comparisons were specifically made. There are letters but they are not explained in the Legend.

4) Is there any blood data from the animal or the human fetuses?

5)Figure 5: Why were the stimulation experiments performed in fibroblasts and not in granulosa or teca cells which are much more important in terms of follicular dynamics and in terms of the origin of the factors analyzed: AMH, AR, DENND1A, etc ? Stimulation of granulosa cells would be very interesting as many of these factors are produced there and not on fibroblasts. This should be clarified in the text as the lack of response in theses experiments most probably is due to the use of the inappropriate cell model. Many of theses factors are crucial for follicle development acting and being produced directly on granulosa cells and not in fibroblasts. This has to be mentioned in the discussion.

Reviewer #2: The work aimed to understand the role of fetal ovarian development in the predisposition to PCOS. The current work described a series of gene expression in different stages of fetal development. However, further study to clarify the interaction network among various ovarian genes are recommended. Moreover, it is too early to get the conclusion about PCOS etiology when the author only observed the expression of PCOS candidate genes. Animal study of PCOS model or even better, the study of women with PCOS are needed.

Reviewer #3: In the present study entitled "Could perturbed fetal development of the ovary contribute to the development of polycystic ovary syndrome in women?, authors have reported that the appearance of PCOS phenotype in adult life could arise by perturbation of expression of PCOS candidate genes during development of the ovary in fetal life. This is a good study and well written. I have some minor comments, as follows:

1. In the title of manuscript, please write "in later life" instead of women.

2. How did Researchers determine the sample size for human and bovine ovaries?

3. Statistical analysis should be written below the tables.

4. The results of this study need more interpretation, and should be compared more with the previous studies.

Reviewer #4: In general, the Manuscript may benefit from several minor revisions, as suggested below:

- All the text needs a language revision by a native English speaker person, in order to improve its readability.

- I would suggest changing conclusions. Study results show that the expression of genes reported related to PCOS by GWAS changes during the development of ovaries in embryos. Nevertheless, no data are reported about an association between the altered expression of these genes and PCOS risk. Therefore, I would suggest avoiding reporting in the conclusion that “These results herald the possibility that the predisposition to PCOS in adult life could arise by perturbation of expression of PCOS candidate genes during development of the ovary” (abstract). This study show that genes related to PCOS change their expression during ovarian development.

In general, the study is very interesting, but it is the first step to understand the role of genes involved in the fetal ovarian development in the predisposition to PCOS. This study shows the pattern of expression of different genes related to PCOS in the different stages of ovarian development, but no data are reported about PCOS, and further studies are required to understand the interaction between ovarian genes and PCOS risk. Therefore, it is not possible to make conclusions such as lines 435-436 and 580-581. I would suggest being softer and reporting these statements in discussion highlighting further lines of research to achieve this objective. I would suggest focusing conclusion of study results and not on something that need to be demonstrated and is supposed. It is important but not demonstrated in this study.

- I would suggest checking the guidelines for Authors of the manuscript and references format.

- Introduction. Lines 90-94. I would suggest improving the discussion about AMH in PCOS. In these patients, AMH seems to lose its usual role as marker of ovarian reserve. In example, to many efforts are spent to identify a correct algorithm which consider woman's age and ovarian reserve markers (AMH) as a tool to optimize the follicle-stimulating hormone starting dose in IVF procedure. Nevertheless, current available evidence regarding PCOS women, particularly the ones with high AMH, suggest that AMH does not seem adequate for this interpretation. I would be glad if the authors discuss this important element, referring to: PMID: 30242498.

- Introduction. Lines 46-59. I would suggest improving the discussion about the role of hyperinsulinemia in PCOS discussing the key role of the use of insulin-sensitizers, such an inositol isoform, gained increasing attention due to their safety profile and effectiveness. Authors may improve this point, taking to account these recent articles: PMID: 30270194; PMID: 31382802.

6. PLOS authors have the option to publish the peer review history of their article (what does this mean?). If published, this will include your full peer review and any attached files.

Reviewer #1: No

Reviewer #2: No

Reviewer #3: No

Reviewer #4: No

---

## [Author Response · Author response to Decision Letter 0]

11 Jan 2020

Reviewer #1: 

The manuscript is very interesting as it addresses the expression of many important markers of PCOS in the ovary in two models: human and bovine. Nevertheless, there are a few important concerns:

1) Clinical data about the mothers and cause of miscarriage is needed. Are there any cases with chromosomopathies involved?

REPLY: The fetuses were obtained from elective abortions and not miscarriages. To clarify we have added the sentence ‘Pregnancies were all terminated for social reasons and all fetuses appeared morphologically normal.’

2) Figure 1: is gestational age a continuous variable here or are the cases grouped ( they seem to represent 3 columns). If the graph represents 2 continuous variables please report de p value for each of the correlations shown. To state that in the human model there is a significant decrease or change in the markers either classify them in groups as with the bovine model and do a statistical comparison or generate a model. Only the R value is given which is not enough.

REPLY: The data have some degree of aggregation, possibly due to the frequency of the surgery in the hospitals where the tissues were obtained from. The P values are found in Table 1 for human data and Table 2 for the bovine data and this is now indicated in the legends of Figures 1 to 4. 

3) Figure 2 H to N please explain what comparisons were specifically made. There are letters but they are not explained in the Legend.

REPLY: We have changed the wording to say ‘Bars with different letters are statistically significantly different from each other (P < 0.05)’ in the legends of Figures 2 to 4. 

4) Is there any blood data from the animal or the human fetuses?

REPLY: Unfortunately not.

5) Figure 5: Why were the stimulation experiments performed in fibroblasts and not in granulosa or theca cells which are much more important in terms of follicular dynamics and in terms of the origin of the factors analyzed: AMH, AR, DENND1A, etc ? Stimulation of granulosa cells would be very interesting as many of these factors are produced there and not on fibroblasts. This should be clarified in the text as the lack of response in theses experiments most probably is due to the use of the inappropriate cell model. Many of theses factors are crucial for follicle development acting and being produced directly on granulosa cells and not in fibroblasts. This has to be mentioned in the discussion.

REPLY: We beg to disagree with some of these comments. In the early stages of fetal ovary development there are no granulosa cells, as follicles have not been formed, and certainly no theca cells they only develop at the antral stage of follicle development (PLoS One 2013, 8: e55578.). Thus it would not be possible to conduct such experiments at early stages of ovary development. Also the AR is expressed in the fibroblasts as these early stages (J Clin Endocrinol Metab. 2011, 96:1754). Many of the factors chosen (e.g. FGF 2, 7 and 9; connective tissue growth factor; TGFβ; platelet-derived growth factor) are active on fibroblasts. Some of the other factors (such as estradiol and androgen) we tested yes can work on granulosa cells but they also work on fibroblasts of the ovary. 

Reviewer #2: 

The work aimed to understand the role of fetal ovarian development in the predisposition to PCOS. The current work described a series of gene expression in different stages of fetal development. However, further study to clarify the interaction network among various ovarian genes are recommended. Moreover, it is too early to get the conclusion about PCOS etiology when the author only observed the expression of PCOS candidate genes. Animal study of PCOS model or even better, the study of women with PCOS are needed.

REPLY: Note that that it is not ethically or practically possible to sample ovarian tissues from human fetuses, allow them to be born and grow up to see if they develop PCOS in later life. You just cannot do it. The current animal models only operate when treating animal late in gestation, well after the time when many of the PCOS genes have already been activated. We agree the next study is to look at how the PCOS candidate genes might be regulated but that is clearly outside the scope of this current study that has already a substantial amount of work presented.

Reviewer #3: 

In the present study entitled "Could perturbed fetal development of the ovary contribute to the development of polycystic ovary syndrome in women?, authors have reported that the appearance of PCOS phenotype in adult life could arise by perturbation of expression of PCOS candidate genes during development of the ovary in fetal life. This is a good study and well written. I have some minor comments, as follows:

1. In the title of manuscript, please write "in later life" instead of women.

REPLY: Done.

2. How did Researchers determine the sample size for human and bovine ovaries?

REPLY: Previous studies have successfully identified changes in gene expression across bovine and human gestation (Dev Biol. 2008, 314:189; FASEB J 2011, 25: 2256) and so we chose similar numbers for the current study. This proved successful in identifying statistically significant relationships.

3. Statistical analysis should be written below the tables.

REPLY: We now say ‘Pearson correlation tests.’

4. The results of this study need more interpretation, and should be compared more with the previous studies.

REPLY: Currently in the introduction we say ‘Many human studies have been conducted in an effort to investigate these PCOS candidate genes, particularly DENND1A and its variants [20], TOX3 [21], FSHR [22], LHCGR [23] and INSR [24]. Another PCOS susceptibility loci was identified earlier by case-cohort studies using microsatellite analyses and it is located in an intron of fibrillin 3 (FBN3) [25]’.

In the Discussion we have discussed previous studies of DENND1A in depth when we said 

‘DENND1A encodes DENND1A (DENN domain containing 1A, connecdenn 1) protein which is believed to assist endocytosis [61]. Human DENND1A possesses 7 variants in total (DENND1A.V1-7) as well as 19 predicted variants (DENND1A.X1-19) (https://www.ncbi.nlm.nih.gov/). The function of these variants is still unclear, however, 2 variants, DENND1A.V1 and DENND1A.V2 have been linked to PCOS [62,63]. DENND1A.V1 encodes a 1009 amino acid protein with C-terminal proline-rich domain, whereas DENND1A.V2 encodes a 559 amino acid protein containing three DENN domains and additionally a C-terminal 33 amino acid sequence [62]. A recent study has found that DENND1A.V2 might play an important role in pathophysiology of hyperandrogenaemia associated with PCOS [62] due to several reasons: significantly elevated levels of DENND1A.V2 mRNA and protein in PCOS thecal cells compared to non-PCOS thecal cells; significantly elevated levels of DENND1A.V2 mRNA in urine of PCOS women compared to women with normal cycles; augmentation of CYP17A1 and CYP11A1 gene transcription, as well as increased dehydroepiandrosterone production in response to overexpression of DENND1A.V2 in normal thecal cells [63]. We were unable to detect the expression of DENND1A.V2 in bovine fetal ovaries or adult thecal cells. However, there were elevated mRNA levels of DENND1A in bovine and human fetal ovaries early in development. ’

Published data on the other PCOS genes is relatively limited and cited. We do not wish to add to the discussion of each gene as the focus of the results is on the pattern of expressions and the relationships of genes within each pattern.

What is asked for would suit a review of this area which maybe done subsequently.

Reviewer #4: 

In general, the Manuscript may benefit from several minor revisions, as suggested below:

- All the text needs a language revision by a native English speaker person, in order to improve its readability.

REPLY: We have enlisted assistance to improve the English grammar and readability, particularly in the introduction and discussion.

- I would suggest changing conclusions. Study results show that the expression of genes reported related to PCOS by GWAS changes during the development of ovaries in embryos. Nevertheless, no data are reported about an association between the altered expression of these genes and PCOS risk. Therefore, I would suggest avoiding reporting in the conclusion that “These results herald the possibility that the predisposition to PCOS in adult life could arise by perturbation of expression of PCOS candidate genes during development of the ovary” (abstract). This study show that genes related to PCOS change their expression during ovarian development.

REPLY: Agree. We have modified the concluding statement of the abstract and the discussion to say ‘With so many relationships between PCOS candidate genes during development of the fetal ovary including TGFβ and androgen signalling we suggest that future studies should determine if perturbations of these genes in the fetal ovary can lead to PCOS in later life.’

In general, the study is very interesting, but it is the first step to understand the role of genes involved in the fetal ovarian development in the predisposition to PCOS. This study shows the pattern of expression of different genes related to PCOS in the different stages of ovarian development, but no data are reported about PCOS, and further studies are required to understand the interaction between ovarian genes and PCOS risk. Therefore, it is not possible to make conclusions such as lines 435-436 and 580-581. I would suggest being softer and reporting these statements in discussion highlighting further lines of research to achieve this objective. I would suggest focusing conclusion of study results and not on something that need to be demonstrated and is supposed. It is important but not demonstrated in this study.

REPLY: Agree. We agree and have made changes to these areas and the concluding statement of the abstract. 

Line 435 have been replaced with ‘This study thus identified relationships between the genetic and fetal development of the ovary’.

Lines 580 has been replaced with ‘With so many relationships between PCOS candidate genes during development of the fetal ovary including TGFβ and androgen signalling we suggest that future studies should determine if perturbations of these genes in the fetal ovary can lead to PCOS in later life.’

- I would suggest checking the guidelines for Authors of the manuscript and references format.

REPLY: We have made these changes. 

- Introduction. Lines 90-94. I would suggest improving the discussion about AMH in PCOS. In these patients, AMH seems to lose its usual role as marker of ovarian reserve. In example, to many efforts are spent to identify a correct algorithm which consider woman's age and ovarian reserve markers (AMH) as a tool to optimize the follicle-stimulating hormone starting dose in IVF procedure. Nevertheless, current available evidence regarding PCOS women, particularly the ones with high AMH, suggest that AMH does not seem adequate for this interpretation. I would be glad if the authors discuss this important element, referring to: PMID: 30242498.

REPLY: Yes we agree with your comments about AMH, but feel that this would depart from the major focus of the article. 

- Introduction. Lines 46-59. I would suggest improving the discussion about the role of hyperinsulinemia in PCOS discussing the key role of the use of insulin-sensitizers, such an inositol isoform, gained increasing attention due to their safety profile and effectiveness. Authors may improve this point, taking to account these recent articles: PMID: 30270194; PMID: 31382802.

REPLY: In this introductory paragraph we introduce what PCOS is. We have not mentioned any of current treatments. To introduce just one treatment would be out of context. To introduce them all would distract from the main focus of the article and experiments carried out. This would be a good topic for a review article in the future maybe.

---

## [Decision Letter · Decision Letter 1]

5 Feb 2020

Could perturbed fetal development of the ovary contribute to the development of polycystic ovary syndrome in later life?

PONE-D-19-32933R1

Dear Dr. Rodgers,

We are pleased to inform you that your manuscript has been judged scientifically suitable for publication and will be formally accepted for publication once it complies with all outstanding technical requirements.

With kind regards,

Simone Garzon

Academic Editor

PLOS ONE

Additional Editor Comments (optional):

Reviewers' comments:

Reviewer's Responses to Questions

**Comments to the Author**

1. If the authors have adequately addressed your comments raised in a previous round of review and you feel that this manuscript is now acceptable for publication, you may indicate that here to bypass the “Comments to the Author” section, enter your conflict of interest statement in the “Confidential to Editor” section, and submit your "Accept" recommendation.

Reviewer #1: All comments have been addressed

Reviewer #3: All comments have been addressed

Reviewer #4: All comments have been addressed

2. Is the manuscript technically sound, and do the data support the conclusions?

Reviewer #1: Yes

Reviewer #3: Yes

Reviewer #4: Yes

3. Has the statistical analysis been performed appropriately and rigorously? 

Reviewer #1: Yes

Reviewer #3: Yes

Reviewer #4: Yes

4. Have the authors made all data underlying the findings in their manuscript fully available?

Reviewer #1: Yes

Reviewer #3: Yes

Reviewer #4: Yes

5. Is the manuscript presented in an intelligible fashion and written in standard English?

Reviewer #1: Yes

Reviewer #3: Yes

Reviewer #4: Yes

6. Review Comments to the Author

Reviewer #1: This questions have been answered properly. The paper now complies with this reviewer´s expectations so I have no further comments.

Reviewer #3: In the present study entitled "Could perturbed fetal development of the ovary contribute to the development of polycystic ovary syndrome in women?, authors have reported that the appearance of PCOS phenotype in adult life could arise by perturbation of expression of PCOS candidate genes during development of the ovary in fetal life. This is a good study and well written. I have some minor comments, as follows:

1. In the title of manuscript, please write "in later lif" instead of women.

2. How did Researchers determine the sample size for human and bovine ovaries?

3. Statistical analysis should be written below the tables.

4. The results of this study need more interpretation, and should be compared more with the previous studies.

*** ALL COMMENTS HAVE BEEN ADDRESSED.

Reviewer #4: Authors have performed the required changes, improving significantly the quality of the manuscript.

I have no further concerns.

7. PLOS authors have the option to publish the peer review history of their article (what does this mean?). If published, this will include your full peer review and any attached files.

Reviewer #1: No

Reviewer #3: No

Reviewer #4: No

---

## [Editor Report · Acceptance letter]

11 Feb 2020

PONE-D-19-32933R1 

Could perturbed fetal development of the ovary contribute to the development of polycystic ovary syndrome in later life? 

Dear Dr. Rodgers:

I am pleased to inform you that your manuscript has been deemed suitable for publication in PLOS ONE. Congratulations! Your manuscript is now with our production department. 

With kind regards,

on behalf of

Dr. Simone Garzon 

Academic Editor

PLOS ONE